# Chemical, Radiometric and Mechanical Characterization of Commercial Polymeric Films for Greenhouse Applications

**DOI:** 10.3390/ma15165532

**Published:** 2022-08-11

**Authors:** John Eloy Franco, Jesús Antonio Rodríguez-Arroyo, Isabel María Ortiz, Pedro José Sánchez-Soto, Eduardo Garzón, María Teresa Lao

**Affiliations:** 1Faculty of Technical Education for Development, Catholic University of Santiago de Guayaquil, Av. C. J. Arosemena Km. 1.5, Guayaquil 09014671, Ecuador; 2Agronomy Department, CIAMBITAL, Agrifood Campus of International Excellence ceiA3, University of Almeria, 04120 Almeria, Spain; 3Department of Mathematics, University of Almería, La Cañada de San Urbano s/n, 04120 Almeria, Spain; 4Institute of Materials Science of Sevilla (ICMS), Joint Center of the Spanish National Research Council (CSIC) and University of Sevilla, 41092 Sevilla, Spain; 5Department of Engineering, University of Almería, La Cañada de San Urbano s/n, 04120 Almeria, Spain

**Keywords:** polymeric films, polyethylene, thermoplastics, cool plastic, greenhouses, multivariate statistical analysis, climate

## Abstract

In the agricultural sector, companies involved in the production of plastic greenhouses are currently searching for a suitable covering adapted for every climate in the world. For this purpose, this research work has determined the chemical, radiometric and mechanical properties of 53 polymeric films samples from Europe and South America. The chemical tests carried out with these samples were elemental analysis (C, H and N) and FT–IR spectrometry. The radiometric properties here studied were the transmission, absorption and reflection coefficients along the spectrum between 300 and 1100 nm. For the mechanical properties, tensile strength, tear strength and dart impact strength, tests were carried out. Finally, all these data were collected, and a multivariate statistical analysis was carried out using the SPSS statistical to group the samples into statistical groups adapted to specific climatic regions. The elemental analysis and FT–IR spectrometry allowed group the samples into nine groups. The samples were grouped according to their chemical (elemental analysis), radiometric and mechanical properties by multivariate analysis. The dendrogram separated five very different groups in terms of number of samples. These groups have specific chemical, radiometric and mechanical characteristics that separate them from the rest. These groups make it possible to narrow down the applications and correlate with the radiometric properties to see in which geographical area of the world they are most effective in increasing yields and achieving higher quality production.

## 1. Introduction

In 2016, plastic covered greenhouses (PCG) reached a total area of 3.019 million Ha worldwide. They are mainly located in Europe, North Africa, the Middle East and China [1]. In Almería there is an area under greenhouse cultivation of 31.034 ha. This production is concentrated in Campo de Dalías, Campo de Níjar and Bajo Andarax [2]. Intensive horticulture under plastic in the area has increased exponentially in the last 50 years [1,3].

Plastic materials used as greenhouse coverings can be classified into flexible films, rigid sheets and nets, although the use of the former far outweighs the other two options [4].

From a climatic point of view, temperature has a great influence on plant growth and metabolism; there is no physiological process that is not influenced by temperature [5]. As far as most agricultural crops are concerned, optimal growth and development is found within the temperature range of 10 and 35 °C [6]. The problem of low temperatures affects crops both in cold climates and in Mediterranean climates with cool winters. In both cases, the problem has been alleviated by the use of thermal plastics, although in cold latitudes, greenhouses are usually heated. These materials absorb or reflect mid-infrared radiation (2500–50,000 nm), especially “atmospheric” (7000–14,000 nm) and prevent radiation emitted during the night from leaving the greenhouse. To manufacture thermal films based on polyolefins, mineral fillers (silica, silicates or borates) are added or the percentage of vinyl acetate is increased [7].

Over the years, mineral fillers used to increase thermicity, as well as low-density polyethylene and the copolymers of ethylene and vinyl acetate/butyl acrylate have evolved. During the 1970s, silica and silicates, especially hydrated aluminium silicates, were mainly used as additives. Over time, the extrusion temperatures used by the converting industry increased and the use of aluminium hydroxide was abandoned, as it decomposes above 180 °C. In the 1990s, the most commonly used fillers were silicates, especially calcined kaolin, but it was found to accelerate film photodegradation, increase turbidity and decrease light transmission. Recently, a new family of mineral fillers has been developed that are non-degrading, do not decrease light transmission and confer low or high turbidity to the film, depending on interest. This has led to more efficient polymers, blocking infrared radiation, resulting in a new generation of ultra-thermal films [8]. However, it is important to note that, currently, greenhouse horticultural production presents problems associated with high temperatures inside the greenhouse in spring–summer conditions.

Some authors, such as Bouzo et al. [9], state that one of the disadvantages of using shading techniques is that the materials currently available generate a non-selective reduction of solar radiation, reducing both near infrared radiation (NIR) (700–200 nm) that is not absorbed by plants and photosynthetically active radiation (PAR). Together with the other forms of cooling, the right choice of cover material can help to regulate greenhouse temperature in tropical and desert areas [7]. Among canopy materials, there is a special interest in cool films, which are photoselective materials, obtained by additives that block the NIR and thus prevent diurnal overheating of the greenhouse [10]. NIR is absorbed during the day by the greenhouse floor, installations and construction elements and during the night it is released back into the greenhouse air as convected heat which increases the greenhouse air temperature [11]. Consequently, NIR is the main source of energy load that must be removed from the greenhouse air to avoid excess heat. NIR radiation can be reduced inside the greenhouse through reflection (metallic pigments) or absorption and interference (pigments with a thin layer of metal oxides) of polymeric films [12]. Reflection and interference are more efficient [10], as the unnecessary energy is reflected outside the greenhouse, whereas through absorption, the covering material will only be able to emit part of the energy outside the greenhouse while another part will be emitted inside the greenhouse, contributing to its warming.

The spectral distribution of global solar radiation, on a horizontal surface, incident on transmitted to a greenhouse can be divided into ultraviolet radiation (UV: 200–400 nm, approximately 5% of the global solar radiation), visible light or photosynthetically active radiation (PAR: 400–700 nm, approximately 45%) and near infrared radiation (NIR: 700–2500 nm, approximately 50%) [11]. In the whole spectrum of global solar radiation, only PAR is absorbed by plants for photosynthesis and is therefore important for their growth [13]. Therefore, desirable canopies for greenhouses in warm, sunny regions should transmit PAR and reject NIR and UV. The contribution of UV radiation to the greenhouse heating load is negligible because it represents only 5% of the global radiation [11]. However, UV rays, especially UVB and UVC, should be rejected because they generate oxidative stress and can damage crops, as well as increase the population of insects, fungi and diseases [11]. In recent years, most of the polymeric films used to cover greenhouses are UV absorbing. Regarding the application of cold plastics in agriculture, García-Alonso et al. [14] found a significant increase of 26% in the yield of pepper crops compared to a commonly used plastic system in the area of Murcia (Spain). In addition, López-Marín et al. [15] recommend the use of photoselective plastics for pepper production in south-western Spain. However, it is important to highlight that there is still a lack of research to develop quality photoselective plastics, as Meca-Abad [16] found that cold plastics generate a significant reduction in PAR radiation (15%), with 38% reductions in NIR radiation.

Another strategy that is proving to be very successful is copolymerisation, which allows the incorporation of several layers, each of which incorporates different properties into the polymer. This allows the technological properties of the polymer to be improved [17].

However, there are still monolayer thermoplastic films that are 200 µm thick and also have significantly lower thermal transmittance and mechanical properties than three-layer films and have a short lifetime. In contrast, three-layer films [18,19,20], typically 220 µm thick made of low-density polyethylene (LDPE) and polyvinyl acetate (PVA), have air bubbles trapped in the middle layer (PROSYN POLYAN), which maintain the temperature in the greenhouse efficiently [21] due to their thermal insulating effect. However, three-layer films based on LDPE, produced by Agrofilm SA (Algeria) without air bubbles show superior mechanical properties compared to monolayer films. Even more, five-layer polymeric films have come onto the market thanks to the work of the Ginegar Plastic Product Ltd. group [22,23]. On this point Dilara and Briassoulis [24] showed that the parameters of the plastics manufacturing process that affect the mechanical properties of the film include melt temperature, die parameters, blowing, rise ratio, draw ratio and cooling conditions. Climatic conditions, such as solar radiation, temperature, humidity, rainfall, wind loads and environmental pollution influence the degradation and mechanical properties of LDPE greenhouse covers. Khan and Hamid [25] reported that UV radiation (290–400 nm) can be absorbed by polyethylene, which leads to photodegradation (oxidation) and thus mechanical degradation. Degradation can increase when metal greenhouse frames come into contact with high temperatures during the day and low temperatures at night [24]. Additionally, Dilara and Briassoulis [26] illustrated another group of factors that influence the mechanical properties of greenhouse polyethylene films which are: microclimatic conditions, such as humidity and internal temperatures, biological activity and agrochemicals.

Salem [27] examined the effect of UV radiation on the mechanical properties of LDPE films containing black carbon and titanium dioxide. The results showed that UV radiation changed the elongation and shear stress of the samples. In this regard, Pacini [28] prioritised the importance of keeping the thermoplastic greenhouse cover in good condition and examined the properties of the covers, including total solar radiation permeability, mechanical properties, service life, effect of weathering and homogeneity of film thickness and width. Finally, Shen and Huang [29] developed a simulation model to calculate the distribution of the molecular structure on the inner surface of the greenhouse canopy and to calculate the yield and elasticity [30].

Verlodt and Verschaeren [31] have investigated the development of a new reflective film containing the anti-NIR additive, which has a higher NIR reflection combined with a higher PAR transmission. The performance of this new polymeric film was evaluated in comparison with a standard PE film for application as a greenhouse cover. The results reflected that the PAR transmittance of the developed film showed a high degree of efficiency. However, its efficiency for NIR reflection was lower. The measured PAR transmittance for such polymeric films were around 77–80%, while the NIR reflectance was around 21–26% [23]. However, a deep knowledge of polymeric film for greenhouse applications is necessary, in particular examining their chemical, radiometric and mechanical properties. This investigation was carried out in order to understand the properties of the range of polymer materials used as greenhouse cover films in the current horticultural sector and to establish differentiated groups according to their properties. The present paper reports the first set of results concerning chemical tests (the elemental analysis of C, H and N) and FT–IR spectrometry, radiometric and mechanical properties.

## 2. Materials and Methods

For this study, a set of 57 types of plastics of different origins were considered: long-life polymeric films, thermal, EVA, with UV additives (Hals), (Quenchers), diffusers, UVC absorbers and anti-drip. Of the 57 types of plastics, there were 3 plastics that were used for disinfection but not for greenhouse coverings and another that gave errors in the reading of radiometric properties. So, after this screening, they were not taken into account in the analyses of this work in which we focused on polymeric films used for greenhouse coverings. They have been named P1 to P53.

### 2.1. Study of Chemical Composition

The chemical composition of polymeric films was studied using FT–IR spectroscopy, which provides information on the presence of organic and inorganic functional groups present in the materials under study, and elemental analysis (N, C, S and H).

This study was carried out by infra-red (IR) spectroscopy, which is based on the absorption of IR radiation by the materials. A JASCO FT/IR-6200 (Tokio, Japan) (equipped with an JASCO Inc IRT-5000 microscope) (Easton, PA, USA) covering a wavenumber range from 5000 to 200 cm^−1^ with CsI optics was used to obtain the infra-red (IR) Fourier transform (FT) spectra.

The measurements were taken in vacuum to avoid interference and the alteration of the samples under study. Samples sized 100 mm × 50 mm, cut from the original sample, were used. The radiation transmitted by the samples, expressed as absorbances, was analysed. A number of 64 scans were performed. As a background of the FT–IR spectra, the measurement was made in the absence of the sample.

For the interpretation of the chemical tests obtained, we represent them in a characteristic zone between 2000 cm^−1^ and 300 cm^−1^, in order to carry out a better analysis and assignment of the bands observed in this zone.

The polymeric film samples were weighed on a balance (precision to two decimal places). Two subsamples were established and one of them was used for destructive analysis by elemental analysis (C, H, N and S) and the other for non-destructive analysis using FT–IR spectroscopy.

For the determination of the elemental chemical analysis of Carbon (C), Hydrogen (H), Nitrogen (N) and Sulphur (S), a LECO TruSpec CHN model was used with an Infrared (IR) detector (Mönchengladbach, Germany) for C, H and S, and a conductivity detector for N. An amount of 2 mg of each sample were used. The combustion was carried out from 950 to 1300 °C in oxygen atmosphere. All results were obtained in duplicate and averaged. High purity sulfamethazine was used for calibration of the equipment with accurate results.

### 2.2. Radiometric Properties

The radiometric determinations of transmittance (% T) and reflectance (% R) of each polymeric films were measured by an integrating sphere using a calibrated spectroradiometer (LI-COR 1800, Lincoln, NE, USA) in the range of 300–1100 nm in the physic applied laboratory of the University of Almería (36°490′ N, 2°240′ W). Lamps is a 200 W quartz tungsten halogen type operated at 3150 °K.

The absorbance (A) has been calculated as: %A(λ) = 100 − (%T(λ) + %R(λ))

The average value of the R, A and T parameters was calculated for the different spectral regions of interest for protected cultivation defined by Pérez-Saiz et al. [32]: ultraviolet (UVA) (300–400 nm), photosynthetically active radiation (PAR) (400–700 nm), near infrared (NIR) (700–1100 nm) and total (300–1100 nm) regions.

The agronomic characterization of these polymeric films has been carried out with respect to the values of R, A and T in % in the different spectral regions, presenting the maximum, average and minimum of the set of plastics studied. The correlation between them has been investigated.

### 2.3. Mechanical Properties

To proceed with the tensile test, we used a die with the shape of the type 5 specimen established by the standard [33,34] (Figure 1) and Hounsfield equipment to carry out the test, which was located in the laboratories of the University of Almería.

The parameters used in the test are 5000 N maximum on the force scale and a test speed of 100 mm/min. For each type of polymeric film thickness (in mm and gauges), surface area (mm^2^), Young’s modulus (MPa), yield stress (MPa), elastic strain, maximum stress (MPa), maximum strain, ultimate stress (MPa), ultimate strain, tear strength (N) and impact strength (g) were recorded.

For the tear resistance test, we used TEARING equipment and the necessary test standard [35]. The dart impact test was carried out with the standard [36]. The equipment is a CEAST.

### 2.4. Statistical Analysis

The results obtained in the tests were analysed with the statistical programme SPSS in order to group the different types of polymeric films into homogeneous groups with respect to their characteristics.

Firstly, a classification of the polymeric films was carried out on the basis of their chemical properties (elemental analysis) and the different groups obtained were characterized. Secondly, in addition to the chemical parameters, radiometric and mechanical properties were considered to be a new classification. Additionally, a correlation matrix has been carried out to determine the interdependence of the optical properties between spectral regions.

## 3. Results and Discussion

### 3.1. Chemical Composition

The results of elemental chemical analysis indicated that N contents were below 0.6%. The samples catalogued from P12 to P18 contain half as much C (in %) and H (in %) as the average, where O is probably an important component (Figure 2). Table 1 shows the maximum, minimum an average percentage values of the composition of the studied polymeric films for C (%), H (%) and N (%) and the % of undetermined composition.

It should be remarked that the C content of polymeric films from P12 to P18 ranges from 45 to 52%.

With the data from the FT–IR spectra obtained, in each case, using graphs, such as the representative spectra included in Figure 3, each group will be described below with some comments on the subject, mentioning the difficulty and complexity that may exist in the interpretation of these spectra as they are not pure organic compounds.

The following considerations have been considered for their interpretation [37,38,39]: The bands at 1460–1450, 1360 and 720 cm^−1^ are assigned to single carbon–hydrogen bond C–H groups (polyolefin types); the band at 1730 cm^−1^ to carbonyl–carboxyl C=O double bonds and the bands at 1240 and 1090 cm^−1^ to single carbon–oxygen bond C–O groups. On the other hand, the bands at 1250 plus the band at 1230 cm^−1^ are assigned to ether-type C–O–C groups. The bands at 1530 cm^−1^ plus 1370 cm^−1^ are assigned to N–O groups of nitrogen–oxygen bonds. Bands at 1100–1300 cm^−1^ are assigned to CO groups in ester compounds. Finally, bands at are assigned 1637 cm^−1^ to OH groups and 960 to C–OH groups.

The results of the analysis allowed to establish 9 groups, although with some differences between them. In fact, Figure 3 shows the FT–IR spectra of a sample corresponding to each group. The groups are described as follows:**Group 1:** Corresponds to samples P-1, P-3, P-10, P-11, P-22, P-23 and P-38. In general, these are “polyolefin” type samples, probably resulting from a mixture of polyethylene and polypropylene in variable proportions perhaps a big amount of the first one. It should be noted that the FT–IR spectra (Figure 3a) of these samples are either prone to oxidation or an additive or filler, possessing a carboxyl group has been added (it was associated to a band at 1739 cm^−1^). It is difficult to assess the type of additive. The bands detected at 1466 and 1456 cm^−1^ are characteristic of C–H bonds (in tables 1460 and 1450 cm^−1^), in addition to the one assigned at 729 cm^−1^ which corresponds to the one tabulated at 720 cm^−1^. In a deep study, it was found that polypropylene had large relative proportions to polyethylene in same samples.**Group 2:** Corresponds to several samples P-2, P-5, P-6, P-7, P-8, P-24, P-26, P-27, P-28, P-30, P-32, P-35, P-36, P-37 and P-39. As in the previous group, these samples are considered to be of the “polyolefin” type, probably resulting from a mixture of polyethylene and polypropylene in variable proportions. These samples can also be considered as partially oxidised or an additive with a carboxyl group was added, and with the possible presence of an N-group additive, which is associated with a small IR band at 1530–1531 cm^−1^ (sometimes at 1534 and 1529 cm^−1^) originating from the presence of N–O bonds (Figure 3b). In a deep study it was found that polypropylene had large relative proportions to polyethylene in some samples.**Group 3:** Corresponds to 10 samples P-4, P-9, P-33, P-34, P-41, P-42, P-43, P-50, P-52 and P-53. These are also “polyolefin” type samples, probably resulting from a mixture of polyethylene and polypropylene. In this group, it can be indicated that the FT–IR spectra (Figure 3c) of these samples are compatible with being unoxidized samples (unlike Group 1), since the band of C–O groups does not appear at 1730 cm^−1^. Furthermore, they are compatible with the possible presence of an N-containing additive (N–O bond band) in Group 2, which is associated with a small IR band at 1541 cm^−1^. Bands at 1534, 1536 and 1539 cm^−1^ are observed in this group of samples.**Group 4:** This group is constituted by samples P-12, P-13, P-14, P-15, P-16, P-17 and P-18. These FT–IR spectra (Figure 3d) appear very saturated, with a profusion of bands as can be seen in the figure. In this case, these could be, or at least are compatible with, samples of polymers of the “polyester” type, bands at 1100–1300 cm^−1^, possibly of an aromatic nature, due to the number of bands present in the spectra. In this group is difficult to assess the presence of additives or fillers due to a lot of bands in the spectra. However, the presence of polyester bands can be associated to an additive including this band in.It should not be omitted that in this group we are dealing with polyolefin samples that are partially degraded, possibly with the intervention of oxygen (see Table 1, Others and Figure 2a). In this sense, the elemental analyses (C, H) of these samples grouped here from P-12 to P-18 are of particular interest since this same group differs from the rest of the samples studied. In fact, they constitute a separate group from the rest of the polymeric films in the set of analysed samples in terms of elemental analysis, since they contain C, H and a significant percentage of the remaining component with respect to 100 (Figure 2), which is assumed to be oxygen.One characteristic that has been observed and should be highlighted in this group of samples is the different thickness of the samples. Additionally, they are not so “transparent” (translucent) and some of them even show a certain lattice or network at the external surfaces.**Group 5:** This group includes only three samples designated as P-19, P-20 and P-40. These are “polyolefin” type samples, resulting from a mixture of polyethylene and polypropylene, being polyolefins with a linear chain (Figure 3e). The FT–IR spectra are similar to samples of Group 1 although with fewer bands.**Group 6:** Comprising only two samples that can be grouped here: P-21 and P-25. These are “polyolefin” type samples, resulting from a mixture of polyethylene and polypropylene but partially oxidised. In this case, unlike the group 5 samples, these polyolefins are found with highly branched chains, which is associated with the presence of a multitude of bands around 1400 cm^−1^ (Figure 3f). It is a difference with samples of Groups 1 and 5.**Group 7:** Comprising only two samples: P-29 and P-31. In this case, these are also “polyolefin” type samples, resulting from a mixture of polyethylene and polypropylene. Moreover, the spectra are compatible, once the bands have been assigned, with a certain “polyether” character of the resulting polymer associated with the bands at 1251 and 1236 cm^−1^ (C-O-C ether bands tabulated at 1250 plus the one at 1230 cm^−1^) and with the presence of a compound (additive or filler) possessing an N–O group (bands at 1530 and 1370 cm^−1^) (Figure 3g).**Group 8:** Comprising only three samples: P-44, P-45 and P-46. This group is compatible with being “polyolefin” type samples, resulting from a mixture of polyethylene and polypropylene, although bands attributable to O–H groups also appear (observed in the FT–IR spectra at 1637 and 970–958–986 cm^−1^ and those tabulated at 1637 and 960 cm^−1^) and which, in turn, are compatible with the presence of an alcohol-type additive—for example polyvinyl—bearing in mind that it is a polymer (Figure 3h). In addition, it is important to mention that a band is also detected at 1534 cm^−1^ and another at 1368–1367 cm^−1^ that can be assigned to the presence of compounds with nitro N–O groups, the bands being tabulated at 1530 and 1370 cm^−1^. Possibly, it could be the same kind of additive as found in precedent samples.**Group 9:** This group includes four samples designated P-47, P-48, P-49 and P-51. These are also “polyolefin” type samples, probably resulting from a mixture of polyethylene and polypropylene, as the precedent samples. However, it is difficult to know the relative proportion of polyethylene and polypropylene present these samples. The study of the FT–IR spectra also allows us to establish that the samples of this group are compatible with the presence of compounds with O–H groups in their structure, similar to samples of Group 8, as well as compounds with N–O groups. The bands observed in these samples are close to the tabulations for this bond at 1530 and 1370 cm^−1^ (Figure 3i). There is also compatibility with carboxyl/carbonyl groups according to the assignment of the observed bands close to the tabulations corresponding to C=O bonds at 1730 cm^−1^ and C–O at 1240 and 1090 cm^−1^. However, the type of additive is difficult to assess in a first study.

### 3.2. Radiometric Properties

Radiation reaching the greenhouse canopy can be reflected, absorbed or transmitted. Reflected radiation is not involved in the production process. However, absorbed radiation can be released by the polymeric films in the form of heat. Transmitted radiation is the radiation that passes through the canopy and reaches the growing system. For each type of polymeric films, the percentage of radiation that reflects, absorbs and transmits has been calculated for each of the bands of interest from an agronomic point of view taking into account their applications, obtaining the maximum, minimum and average values for all the samples (Table 2).

In the following figures we can see how the percentage of reflected, transmitted and absorbed radiation varies for each of the 53 types of polymeric films. It is noteworthy that in the UVA spectral region the absorbed percentage is higher than the transmitted percentage for some types of polymeric films (Figure 4a). However, in the PAR (Figure 4b) and NIR (Figure 4c) regions, the transmitted percentage is much higher than the reflected and absorbed ones, which are below 30%, even when considering the spectrum studied as a whole (Figure 4d).

If the mean value of reflectance (Figure 5a), transmittance (Figure 5b) and absorbance (Figure 5c) is that presented by the total spectrum (Figure 4d), the reflectance, transmittance and absorbance corresponding to regions above this line are higher and below are lower. It can be seen that the region with the highest reflectance is blue and the region with the lowest reflectance is UVA. Among the samples studied, there are variations in reflectance of around 20%, except in the UVA region, which is only 14%. The polymeric films with the highest NIR reflectance are P-1, P-21 and P-29 and can therefore be considered cold plastics. However, these samples also have the highest reflectance in the PAR region.

It can be seen that all regions have similar transmittance except for the UV region, which is lower. Among the polymeric films studied, there are variations in transmittance of around 40%, except in the UVA region, which reaches 65%. The polymeric films with the lowest NIR transmittance are P-1, P-21 and P-22 and can therefore be considered cold polymers. However, these plastics also have the lowest transmittance in the PAR region.

It should be noted that the highest absorbance is observed in the UV region. Among the polymeric films studied, there are variations in absorbance of around 20%, except in the UVA region, which reaches 68%.

Table 3, Table 4 and Table 5 show the correlations obtained between reflected, transmitted and absorbed radiation in the different spectral regions. These correlations are significant, positive and very high, which means that the higher the reflectance, transmittance or absorbance of the PAR, the higher the reflectance, transmittance or absorbance of the NIR. This is true for all spectral regions, except for the case of UVA radiation. In the latter case the correlation is not significant with the other spectral regions. This may be due to the use of different additives to modify the behaviour of the polymeric films in this spectral region.

### 3.3. Mechanical Analysis

Table 6 shows the maximum, minimum and average values of the different parameters defining the tensile strength according to the standard [33], the tearing strength (Newton, N) and the impact strength of the darts of all the polymeric films studied here.

We can say that they behave as ductile polymers with a yield point, where it can be seen that they correspond to the type b and c graphs of the visual depiction provided in the standard [33].

### 3.4. Classification of Analysed Polymeric Films

In order to find a possible classification of the polymeric films on the basis of the observed variables, a non-hierarchical cluster analysis of the k-means has been carried out. Since the variables have different units of measurement, the analysis has been performed on the typed variables. The classification obtained in this analysis is presented in Table 7.

For the classification in these groups, all the variables analysed were significant except for nitrogen, maximum stress, breaking strain, tear strength and T_UVA. The variables that were significant are presented in Table 8.

Average values of the variables for each group are presented in Table 9 and Figure 6, where it can be seen that **Group 1** has a high content of carbon and hydrogen. In terms of mechanical properties, it is characterised by a high Young’s modulus. Regarding the radiometric behaviour, it is observed that the reflected radiation is the highest for all spectral regions, and the transmitted radiation is the lowest. This may limit production in certain geographical areas, as the PAR is around 57.08%. In addition, the absorbed radiation is the highest in virtually all spectral regions. These polymers as films would be suitable for tropical areas where temperatures and radiation intensity are very high, as the NIR radiation responsible for heating the interior of the greenhouse is lower. This extreme heating can cause stress to crops.

**Group 2** also has high carbon and hydrogen contents, together with Young’s modulus. However, the maximum deformation and breakage are maximum. Therefore, these are polymeric films that can register important deformations. This property is very important in the handling and placement on the greenhouse structure. In terms of radiometric properties, they have an acceptable reflection, together with a fairly high transmission, which means that absorption is quite low. Therefore, it is a very efficient group that utilises solar radiation for crops. This group as polymeric films would be suitable for areas with a Mediterranean climate where crops are grown from autumn to spring, and it is important that the PAR transmission be high to achieve maximum earliness. In addition, the transmission in the NIR region and in the total is very high, which provides heat in the coldest months of the winter. To alleviate the hot days at the end of summer, techniques, such as liming, are used to lower the extreme temperatures at the beginning of the crop.

**Group 3** is characterised by its low carbon and hydrogen content, which means that a very high percentage of other chemical components are involved in its composition, and its thickness is greater than that of the previous groups. In terms of mechanical behaviour, the values are much lower than Group 2. However, the radiometric behaviour is similar to group 2, although it has the highest absorption in the UVA region.

**Group 4** is similar to Group 3 in terms of chemical composition and mechanical behaviour, except for the dart impact resistance, which is the lowest (110). Additionally, the thickness is the lowest. However, the radiometric properties are quite good with a very high transmission, which makes the absorption and reflection quite low. Therefore, these polymeric plastic as films are very efficient in harnessing solar energy.

Finally, **Group 5** is similar to the two previous groups in terms of chemical composition. In addition, these are thicker polymeric films, which means that the stress and strain at yield, maximum stress and strain at break are the highest, but the strains at break and maximum strains are the lowest. The radiometric behaviour is very similar to Group 3 with high transmissions in all spectral regions, accompanied by low absorbance and reflectance. Therefore, the balance for cultivation is quite favourable. This polymeric film is suitable for geographical areas exposed to high stresses in the Mediterranean zone, although its price is very high due to its high thickness.

## 4. Conclusions

Fifty-three samples of greenhouse-covering polymeric films used in Europe and South America were selected. By studying their elemental chemical composition (C, H and N) and FT–IR spectra, mechanical and radiometric properties, we concluded the following:

The elemental chemical composition and FT–IR spectra have allowed the classification of nine groups with specific characteristics in terms of assignation of bands of these spectra. In addition, there is a Group 4 that has half of C and H in favour of other components. It is also seen that Groups 2 and 3 are the ones with the highest number of samples, followed by 1 and 4. The rest of the groups are very small considering the number of grouped samples.

In terms of radiometry, differential behaviour is observed for transmitted, reflected and absorbed radiation in the polymeric films studied. There is no significant correlation between the radiometric behaviour in the UV region and the other regions studied. However, there is a significant and positive correlation between the other spectral regions studied. As a consequence, with the materials currently used, the application of additives to modify the NIR region generates a modification of the PAR region and, therefore, it affects the efficiency of the coatings for the generation of bioassimilates.

The mechanical behaviour of these polymeric films is of type b and c, which corresponds to ductile plastics with a yield point.

In relation to the classification obtained in this investigation taking into consideration the chemical composition (C, H and N), radiometry and mechanical properties, it has been possible to separate results into five groups. Group 1 would be suitable for tropical climate zones, because it presents low transmittance in NIR spectrum region, which will allow for lower temperatures closer to the optimal range for crop development. However, Group 2 is more suitable for Mediterranean climates, as it allows NIR radiation to pass through, which causes the greenhouse to heat up in autumn/winter and spring, improving the thermal integral of the greenhouse environment and obtaining higher productions with greater quality. The other three groups are split from Group 4 of the classification, taking into consideration their chemical composition. They can be used in Mediterranean climate areas, although they have some additions that give them special characteristics. Group 3 absorbs more UV radiation, which can put the lifespan of the polymer at risk due to its degradation, and Group 4 has very low dart impact and thickness, which can put the integrity of the film at risk due to causes related to the systems for fixing the film to the structure and wind loads. However, its radiometric properties are excellent. Finally Group 5 has higher thickness, which makes the stresses higher and the deformations very low.

A deep study of the presence of other elements, besides C, H and N and possible fillers, will be of great interest for the modification of the classifications deduced in the present investigation. It will be the subject of a future paper.

## Figures and Tables

**Figure 1 materials-15-05532-f001:**
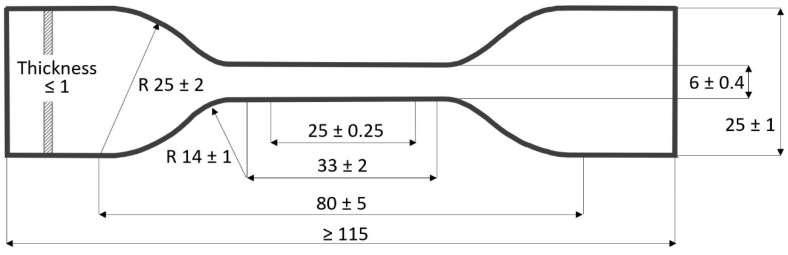
Tensile test specimen (specimen type 5), for non-rigid plastics (measured in mm).

**Figure 2 materials-15-05532-f002:**
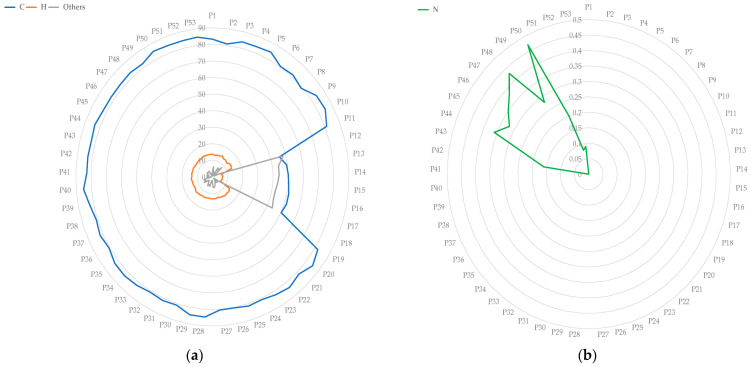
Chemical composition (wt.%) of each sample of polymeric films: (**a**) C, H and others; (**b**) N.

**Figure 3 materials-15-05532-f003:**
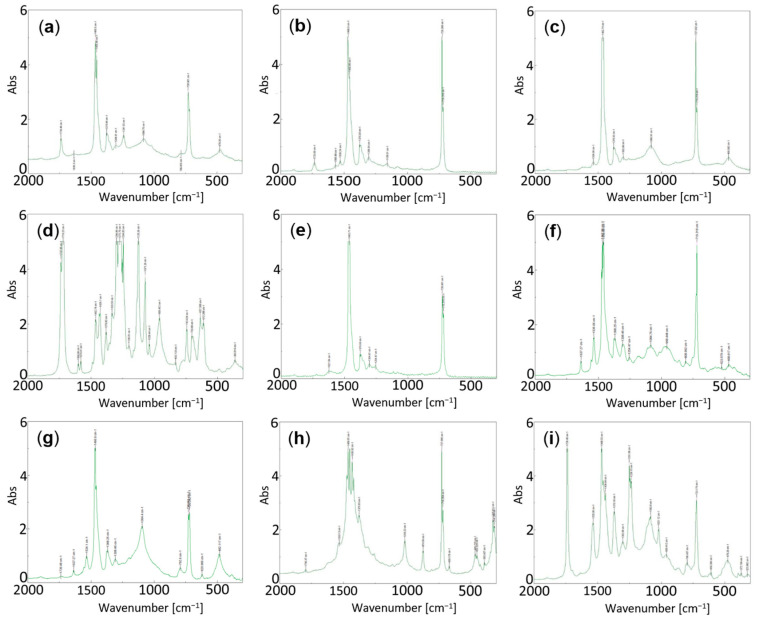
Representative FT-IR spectra of polymeric films: (**a**) P1 (Group 1), (**b**) P5 (Group 2), (**c**) P9 (Group 3), (**d**) P13 (Group 4), (**e**) P20 (Group 5), (**f**) P21 (Group 6), (**g**) P29 (Group 7), (**h**) P45 (Group 8) and (**i**) P49 (Group 9).

**Figure 4 materials-15-05532-f004:**
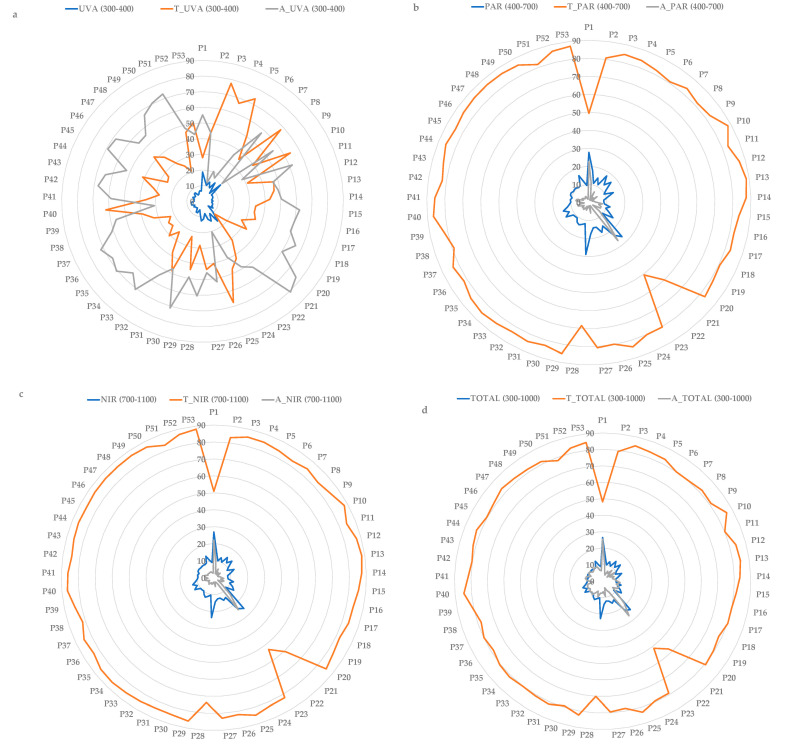
Radiometric behaviour of the polymeric films in the spectral region: (**a**) UVA, (**b**) PAR, (**c**) NIR, (**d**) Total spectrum.

**Figure 5 materials-15-05532-f005:**
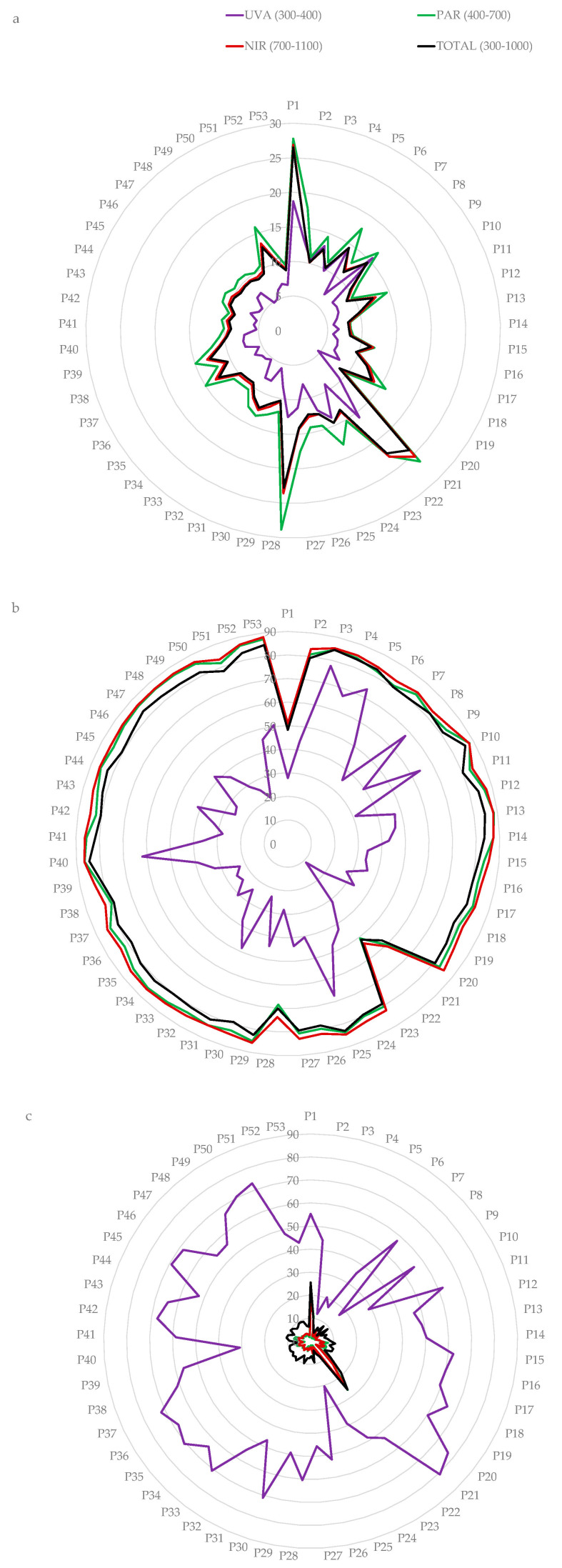
Radiation (**a**) reflected, (**b**) transmitted and (**c**) absorbed in the different spectral regions of the polymeric films.

**Figure 6 materials-15-05532-f006:**
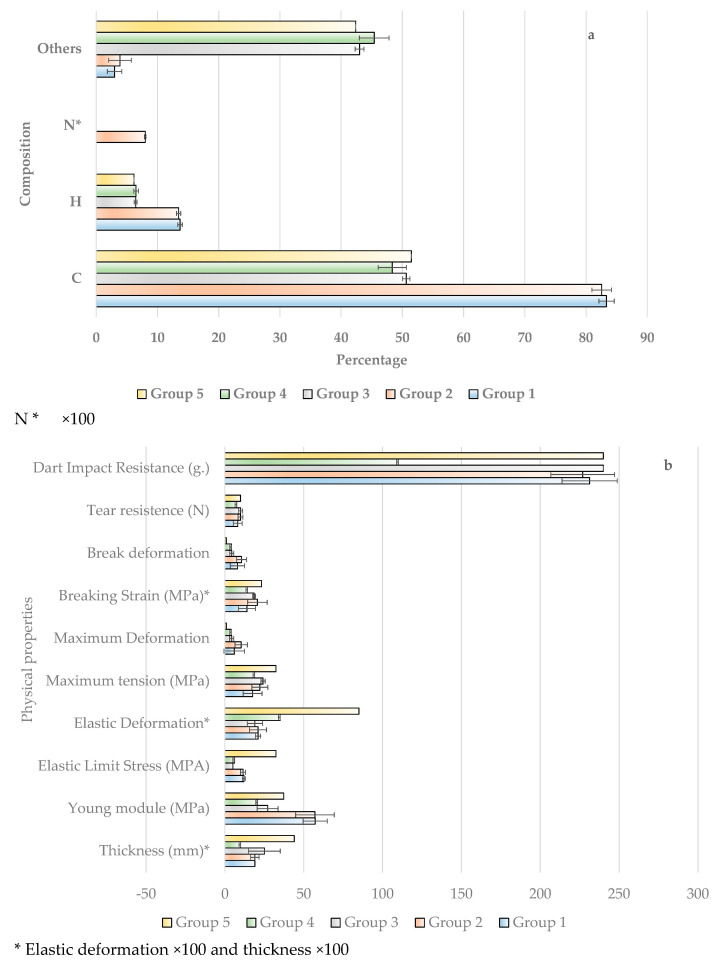
Mean values corresponding to the five classification groups established by multivariate statistical analysis of all (**a**) chemical, (**b**) radiometric and (**c**) mechanical parameters of the polimeric film.

**Table 1 materials-15-05532-t001:** Maximum, minimum and average values of wt.% C, H, N and other elements of the set of analysed polymeric films.

	C (%)	H (%)	N (%)	Others (%)
Max	85.22	14.25	0.47	48.21
Min	45.72	6.07	0.00	0.53
Average	78.28	12.54	0.06	9.10

**Table 2 materials-15-05532-t002:** Values (in %) of maximum, minimum and average of reflectance, transmittance and absorbance of the studied polymeric films.

		UVA	PAR	NIR	Total
Reflectance (%)	Maximum	18.70	28.83	26.86	26.47
	Minimum	4.64	8.35	8.21	8.04
	Average	8.13	13.37	12.10	11.79
Transmittance (%)	Maximum	77.57	88.39	88.21	86.06
	Minimum	11.24	49.48	50.92	48.22
	Average	38.97	81.72	83.09	79.40
Absorbance (%)	Maximum	80.64	26.57	23.76	26.38
	Minimum	12.27	1.49	2.39	3.67
	Average	53.64	5.10	4.95	9.00

**Table 3 materials-15-05532-t003:** Correlations between reflected radiation in different regions.

	R_UVA	R_PAR	R_NIR
R_UVA	1	0.754	0.725
R_PAR	0.754	1	0.979
R_NIR	0.725	0.979	1
R_TOTAL	0.759	0.982	0.999

**Table 4 materials-15-05532-t004:** Correlations between transmitted radiation in different regions.

	T_UVA	T_PAR	T_NIR	T_TOTAL
T_UVA	1	0.287	0.268	0.413
T_PAR	0.287	1	0.993	0.984
T_NIR	0.268	0.993	1	0.987
T_TOTAL	0.413	0.984	0.987	1

**Table 5 materials-15-05532-t005:** Correlations between absorbed radiation in different regions.

	A_UVA	A_PAR	A_NIR	A_TOTAL
A_UVA	1	0.199	0.144	0.458
A_PAR	0.199	1	0.986	0.941
A_NIR	0.144	0.986	1	0.939
A_TOTAL	0.458	0.941	0.939	1

**Table 6 materials-15-05532-t006:** Maximum, minimum and average values of the different parameters defining tensile strength according to the standard [33], tear resistance (N) and dart impact resistance of all the polymeric films studied in this work.

	Maximum	Minimum	Average
Thickness (mm)	0.44	0.10	0.19
Young’s modulus (MPa)	75.35	11.96	52.97
Elastic Limit Stress (MPa)	32.39	5.00	11.25
Elastic Deformation	0.85	0.12	0.23
Maximum Tension (MPa)	33.92	11.91	21.89
Maximum Deformation	15.35	0.32	9.08
Breaking Strain (MPa)	33.92	9.83	19.68
Break Deformation	15.35	0.88	9.46
Tear Resistance (N)	11.36	4.00	9.59
Dart Impact Resistance (g)	240.00	110.0	221.60

**Table 7 materials-15-05532-t007:** Number of cases in each cluster.

Groups	Samples
1	4 samples: P-1, P-21, P-22 and P-28
2	42 samples: the rest
3	3 samples: P-15, P-16 and P-18
4	3 samples: P-12, P-13 and P-14
5	1 sample: P-17

**Table 8 materials-15-05532-t008:** Significant variables in group formation.

	Cluster		Error		F	Sig.
	Root Mean Square	Gl	Root Mean Square	Gl		
C	12.762	4	0.020	48	642.207	0.000
H	12.769	4	0.019	48	663.201	0.000
N	0.795	4	1.017	48	0.782	0.543
Others	12.789	4	0.018	48	726.474	0.000
Thickness (mm)	8.552	4	0.371	48	23.071	0.000
Young’s Modulus (MPa)	6.088	4	0.576	48	10.570	0.000
Elastic Limit Stress (MPa)	11.125	4	0.156	48	71.206	0.000
Elastic Deformation	8.849	4	0.346	48	25.581	0.000
Maximum Tension (MPa)	2.127	4	0.906	48	2.348	0.068
Maximum Deformation	4.009	4	0.749	48	5.350	0.001
Breaking Strain (MPa)	1.841	4	0.930	48	1.980	0.113
Break Deformation	5.298	4	0.642	48	8.253	0.000
Tear Resistance (N)	2.110	4	0.907	48	2.325	0.070
Dart Impact Resistance (g)	8.966	4	0.336	48	26.676	0.000
R_UVA	4.050	4	0.746	48	5.429	0.001
R_PAR	9.393	4	0.301	48	31.250	0.000
R_NIR	10.593	4	0.201	48	52.800	0.000
R_TOTAL	10.366	4	0.219	48	47.227	0.000
T_UVA	1.413	4	0.966	48	1.463	0.228
T_PAR	11.136	4	0.155	48	71.678	0.000
T_NIR	11.028	4	0.164	48	67.113	0.000
T_TOTAL	10.874	4	0.177	48	61.394	0.000

**Table 9 materials-15-05532-t009:** Mean values corresponding to the five classification groups established by multivariate analysis of all chemical, radiometric and mechanical parameters of the polymeric films (Not Sig. = Not significant).

	Group					
	1	2	3	4	5	
	Average	Average	Average	Average	Average	
C	83.34	82.55	50.64	48.36	51.48	
H	13.68	13.45	6.43	6.50	6.15	
N *	0.00	0.08	0.00	0.00	0.00	Not sig.
Others	2.98	3.88	43.01	45.40	42.37	
Thickness (mm)	0.19	0.19	0.25	0.10	0.44	
Young’s Modulus (MPa)	57.25	57.10	27.10	20.63	37.24	
Elastic Limit Stress (MPa)	12.04	11.48	5.05	6.13	32.39	
Elastic Deformation	0.21	0.21	0.19	0.35	0.85	
Maximum Tension (MPa)	17.55	22.11	24.18	18.86	32.39	Not sig.
Maximum Deformation	5.91	10.29	4.31	4.03	0.85	
Breaking Strain (MPa) *	13.97	20.61	18.49	14.31	23.17	Not sig.
Break Deformation	7.92	10.56	4.35	4.10	0.88	
Tear Resistance ((N) *	8.11	9.85	9.94	7.43	9.87	Not sig.
Dart Impact Resistance (g)	231.25	226.90	240.00	110.00	240.00	
R_UVA	14.13	7.85	6.33	6.36	6.74	
R_B	27.21	13.60	11.02	9.23	9,.92	
R_R	25.41	11.70	10.96	8.92	9.89	
R_FR	23.80	10.97	10.95	8.87	9.91	
R_PAR	26.46	12.68	11.04	9,07	9.93	
R_NIR	24.70	11.27	10.86	8.89	9.85	
R_TOTAL	23.92	11.01	10.51	8.69	9.61	
T_UVA	24.65	40.46	31.86	45.44	35.67	Not sig.
T_B	54,38	81.30	79.43	87.04	81.52	
T_R	59.10	85.02	84.07	87.68	83.88	
T_FR	59.81	85.39	85.09	87.84	84.55	
T_PAR	57.08	83.59	82.19	87.47	82,92	
T_NIR	59.70	84.91	83.98	87,62	83.92	
T_TOTAL	56.73	81.17	79.89	84.02	80.28	
A_UVA	62.28	52.39	62.75	48.64	59.29	
A_B	18.41	5.10	9.55	3.73	8.56	
A_R	15.49	3.28	4.97	3.40	6.23	
A_FR	16.39	3.81	3.96	3.29	5.54	
A_PAR	16.47	3.96	6.77	3.46	7.15	
A_NIR	15.60	4.00	5.15	3.49	6.22	
A_TOTAL	19.57	8.02	9.76	7.33	10.35	

Note: * Parameters not significant (*p* < 0.05) for the establishment of clusters.

## Data Availability

Not applicable.

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
