# Peer review of "Chemical, Radiometric and Mechanical Characterization of Commercial Polymeric Films for Greenhouse Applications"

_materials, 2022, doi:10.3390/ma15165532_

Round 1
Reviewer 1 Report
The manuscript entitled “Chemical, Radiometric and Mechanical Characterisation of Commercial Plastic Films for Greenhouse Applications” describes the commercially available polymer films, namely 53 of them. This paper has several problematic parts which makes the manuscript not suitable for publication. The paper is confusing.
First of all, what is the point of this research? The authors have studied the properties of commercially available which were characterised in the terms of mechanical properties before they were commercialized. In addition, the authors did not present what kinds (samples) of films are studied at all, since they have named them from P1 to P53, and stated there are several of them (lines 153-159). If we buy a polymeric film, we will know is it based on PE or PP, or polyester, etc. my question is why haven’t you named exact what kind of film you have studied? All samples should be summarized in one table and described with their most important properties and producers. In addition, the density, grammage, thickness, are the parameters often given by the producers which in the end can influence the basic mechanical properties. The authors used 53 polymer films and grouped them without any credible measurable value. The results based on that are not credible. In conclusion, the authors have stated that for example the group 1 is suitable for tropical climatic zones – my question is what we have with this information (and with other when “group” is used) when we do not know what is behind/or in that group!
The manuscript has numerus flaws related to the FTIR characterization. All the groups given by the authors have been defined as polyolefin type whit some slight small differences. This are all only assumptions which are not confirmed by any other reference or method. For example, lines 297-313, the authors talk about polyesters but they did not provide any assignation related to ester bond. This is only one example. In line 316, the authors talak about relative purity of the films, based on what this is concluded. Missing the technique of FTIR, ATR or KBr? Moreover, how can you say with 100% accuracy that you got real parameters since you haven't said anything about the baseline corrections and adjustments? Please explain. The presented spectra are very complexed and cannot be interpreted in a way that authors did it.
There are missing information related to of the results. Only some results are presented, not for all samples. Why only the table 3 presents the results of P-16?
Lines 211: what do you mean by the “agronomic characterization”
Line 194-195: How was integrating sphere used for the transmittance measurements since it measured diffuse reflections?
Figures 1, 2 and 4 are totally unnecessary.
The authors in the manuscript often used plastic films instead of polymeric films.
How can authors state the average values of different films in table 4 and 8?
Author Response
The manuscript entitled “Chemical, Radiometric and Mechanical Characterisation of Commercial Plastic Films for Greenhouse Applications” describes the commercially available polymer films, namely 53 of them. This paper has several problematic parts which makes the manuscript not suitable for publication. The paper is confusing.
First of all, what is the point of this research? The authors have studied the properties of commercially available which were characterised in the terms of mechanical properties before they were commercialized. In addition, the authors did not present what kinds (samples) of films are studied at all, since they have named them from P1 to P53, and stated there are several of them (lines 153-159). If we buy a polymeric film, we will know is it based on PE or PP, or polyester, etc. my question is why haven’t you named exact what kind of film you have studied? All samples should be summarized in one table and described with their most important properties and producers. In addition, the density, grammage, thickness, are the parameters often given by the producers which in the end can influence the basic mechanical properties. The authors used 53 polymer films and grouped them without any credible measurable value. The results based on that are not credible. In conclusion, the authors have stated that for example the group 1 is suitable for tropical climatic zones – my question is what we have with this information (and with other when “group” is used) when we do not know what is behind/or in that group!
Response: Thank you very much for your comments. The authors consider that the paper is not confusing. It is of interest and within the overall scope of “Materials”. Companies involved in the production of different polymeric materials as films, for greenhouse applications, are currently searching for a suitable covering adapted for every climate in the world. The authors consider that a deep knowledge of this kind of materials is necessary, in particular examining their chemical, radiometric and mechanical properties, as described in the submitted paper. Thus, this research work has been carried out in order to know these properties in a set of commercial polymer materials (53 samples from Europe and South America) used as greenhouse cover films in the current horticultural sector. As a first approximation, it was intended to establish differentiated groups according to their properties. It was mentioned at the end of the Introduction section.
It can be emphasized that this work was performed under a research project, named “Transfer Project”, with a private company, as included in “Funding”. Thus, part of this project is confidential (properties and producers of the samples, for instance) according to the conditions of this “Transfer Project”. The polymeric films have been selected in a first step by the company to study their properties, making the experimental measurements at the laboratory under the conditions described in the paper. A short description was included in the submitted paper (lines 153-159). A Table of the most important properties and producers is not available. The polymeric materials used for this investigation were purchased and they have not been pre-treated. They are under pristine condition for the measurements.
The chemical tests were elemental analysis (C, H and N) and FT-IR spectrometry, techniques available outside the laboratories of University of Almería. These test were performed at the Institute of Materials Science of Sevilla (ICMS) under a collaborative work. The radiometric properties, determined in this research at the University of Almería, were the transmission, absorption and reflection coefficients along the spectrum between 300 and 1100 nm. The mechanical properties, tensile strength, tear strength and dart impact strength tests, were carried out at the University of Almería. All these data were collected, and a multivariate analysis was carried out to group the samples into statistical groups with the aim of being adapted to specific climatic regions. A first insight into the properties of these polymeric films was reported. Next step will be a more extensive study of additional properties.
According to the results of this investigation under the “Transfer Project”, in part presented and reported in the paper, it is possible to narrow down the applications and correlate with the radiometric properties to see in which world geographical area of the world these plastic films are most effective in increasing yields and achieving higher quality production. The authors consider that the paper is original, not being submitted in other journals and it will be of interest for the readers of “Materials”. It was confirmed by other revisions.
The manuscript has numerous flaws related to the FTIR characterization. All the groups given by the authors have been defined as polyolefin type whit some slight small differences. This are all only assumptions which are not confirmed by any other reference or method. For example, lines 297-313, the authors talk about polyesters but they did not provide any assignation related to ester bond. This is only one example. In line 316, the authors talak about relative purity of the films, based on what this is concluded. Missing the technique of FTIR, ATR or KBr? Moreover, how can you say with 100% accuracy that you got real parameters since you haven't said anything about the baseline corrections and adjustments? Please explain. The presented spectra are very complexed and cannot be interpreted in a way that authors did it.
Response: The FTIR provided information about the bonds present in the 53 polymeric films according to the absorption of IR radiation by the polymeric films. In general, the spectra are complex and, as a first approximation, an evaluation of the assignation of main bands associated to functional groups was performed. It was mentioned in the work the difficulty and complexity that may exist in the interpretation of these spectra. The original film samples (sizes of ~100 x 50 mm) were used as received for the obtention of the FTIR spectra without any preparation or pre-treatment. Thus, the technique of KBr pellet, with previous grinding of KBr and sample, mixing and pressing to prepare a pellet, was avoided. It has been mentioned in the revised version. A number of 64 scans were made for each sample and the FTIR spectra were obtained in vacuum.
Corrections of the background of the FTIR spectra was made by measurement in the absence of sample. Baseline corrections to zero absorbance and adjustments were performed. Some differences were observed by FTIR, although the interpretation was difficult and other methods were not used for confirmation. These methods were not assumed by funding under the conditions of the Transfer Project. The Ultraviolet visible spectra (UV-vis) of these samples were investigated. In fact, the experimental conditions were included in the work (section 2.1., being deleted in the revised version). The interpretation of these results was difficult. However, with the FTIR information, it was possible an attempt to classify the samples in 9 Groups, being a first approximation. The revised version includes some additional assignation, as proposed by the reviewer.
There are missing information related to of the results. Only some results are presented, not for all samples. Why only the table 3 presents the results of P-16?
Response: Results of sample P-16: Table 3 is presented as an example to check the elemental chemical analysis performed in this work.
Lines 211: what do you mean by the “agronomic characterization”
Response: The agronomic characterization is defined by interesting regions of spectrum related to horticultural production as UVA (Physiological stress), PAR (Photosynthetic Active Radiation) and NIR (Energetic balance of the greenhouse).
Line 194-195: How was integrating sphere used for the transmittance measurements since it measured diffuse reflections?
Response: This methodology allows precision measurements to be made with a known light solar source (with an emission radiometric spectrum like that of the sun), which passes through the polymer and, due to the reflection capacity of the integrating sphere, the radiometric detector receives and quantifies all the radiation transmitted by the polymer, both direct and diffuse.
Figures 1, 2 and 4 are totally unnecessary.
Response: Figures 1, 2 and 4 have been deleted in the revised version.
The authors in the manuscript often used plastic films instead of polymeric films.
Response: Thank you very much for your appreciation. This correction has been incorporated in te revised version along the text.
How can authors state the average values of different films in table 4 and 8?
Response: The average values have been calculated in Tables 4 and 8…
Reviewer 2 Report
The paper contains a valuable view of chemical, radiometric and mechanical characterization of commercial plastic films for greenhouse applications. The authors investigated an interesting topic and the objective of the paper is of worldwide interest and fits well within the overall scope of the journal.
Author Response
The paper contains a valuable view of chemical, radiometric and mechanical characterization of commercial plastic films for greenhouse applications. The authors investigated an interesting topic and the objective of the paper is of worldwide interest and fits well within the overall scope of the journal.
Response: Thank you very much for your evaluation concerning the original manuscript submitted for revision and publication to “Materials”. Your are right: the objective of this investigation after publication is of worldwide interest.
Reviewer 3 Report
Journal: Materials (ISSN 1996-1944).
Manuscript ID: materials-1814727
Type: Article
Title: Chemical, Radiometric and Mechanical Characterisation of Commercial Plastic Films for Greenhouse Applications.
Authors: John E. Franco, Jesús A. Rodríguez-Arroyo, Isabel. M. Ortiz, Pedro J. Sánchez-Soto, Eduardo Garzón, María Teresa Lao.
a) In the Title: revise “Characterization”.
b) Introduction … radiation (2,500-50,000 nm), especially "atmospheric" (7,000-14,000 nm) revise … (2500-50000) nm and (7000-14000) nm ad all like this.
c) Why the author didn’t measure the thermal conductivity, hardness, and flexural strength of the material?
d) Refer to these refs. Very useful for other mechanical measurements of the materials
DOI: https://doi.org/10.1088/1742-6596/1795/1/012059
DOI: https://doi.org/10.1007/s10854-017-6660-9
Author Response
a) In the Title: revise “Characterization”.
Response: It was revised and changed in the title.
b) Introduction … radiation (2,500-50,000 nm), especially "atmospheric" (7,000-14,000 nm) revise … (2500-50000) nm and (7000-14000) nm ad all like this.
Response: It was amended in this part of the Introduction section.
c) Why the author didn’t measure the thermal conductivity, hardness, and flexural strength of the material?
Response: This comment is interesting. However, equipments and facilities for these measurements were not available when the experimental work was performed. The authors plan to make these measurements in future studies under a collaboration.
d) Refer to these refs. Very useful for other mechanical measurements of the materials
DOI: https://doi.org/10.1088/1742-6596/1795/1/012059
DOI: https://doi.org/10.1007/s10854-017-6660-9
Reviewer 4 Report
The manuscript discusses the various characteristics of the commercially available plastic films used in greenhouse applications and the grouping of those. Though the topic of discussion is interesting, detailed discussions are missing and hence it seems more like a scientific report. The authors should address the following points in the revised manuscript.
1. You have to mention the initial conditions of the plastics that are subjected to the various characterization studies. For example, are they newly purchased or not? Whether any initial pre-treatment is done? etc
2. Why the elemental analysis is restricted only to C, N and H? What about the role of other additives or fillers? Grouping only on the basis of C, N and H may be inaccurate and will lead to wrong conclusions.
3. Fig.6 is of low resolution and is difficult to infer. Moreover, similar to elemental analysis, I felt FTIR studies were also limited.
4. You need to mention the intensity of the radiation that you have used in the radiometric test, the duration and so on. Post characterization of the radiation-exposed samples may also provide valuable insights.
5. You need to correlate the mechanical characterization results with the elemental analysis and FTIR. Since the results obtained via elemental analysis and FTIR are limited, the correlation that you have shown in the manuscript may be improper or inaccurate.
Author Response
The manuscript discusses the various characteristics of the commercially available plastic films used in greenhouse applications and the grouping of those. Though the topic of discussion is interesting, detailed discussions are missing and hence it seems more like a scientific report. The authors should address the following points in the revised manuscript.
- You have to mention the initial conditions of the plastics that are subjected to the various characterization studies. For example, are they newly purchased or not? Whether any initial pre-treatment is done? Etc
Response: The polymeric materials used for this investigation are newly purchased. They have not been pre-treated. They are under pristine condition.
- Why the elemental analysis is restricted only to C, N and H? What about the role of other additives or fillers? Grouping only on the basis of C, N and H may be inaccurate and will lead to wrong conclusions.
Response: The elemental analyzer available for this study is a LECO equipment TruSpec CHN model using combustion from 950 to 1300 ºC in oxygen atmosphere. Then, it is possible to analyze Carbon, Hydrogen, Nitrogen and other elements, such as S. This equipment use an Infrared detector for C, H and S and a conductivity detector for N using 2 mg of sample. S was not detected in the samples or its content was under the detection limit of this equipment.
The additives or fillers were not investigated because, as a first approximation, the elemental or basic analysis of C, H and N was considered. Other equipments for analytical measurements were not available in the investigation (under the conditions of Funding by a Transfer Project with a Company). Then, the conclusions of the study are based in the analytical results of only these elements, being the majority content (in percentages) of the polymeric samples.
- Fig.6 is of low resolution and is difficult to infer. Moreover, similar to elemental analysis, I felt FTIR studies were also limited.
Response: The authors modified the resolution of this Figure.
- You need to mention the intensity of the radiation that you have used in the radiometric test, the duration and so on. Post characterization of the radiation-exposed samples may also provide valuable insights.
Response: The intensity of the radiation … Experimental conditions have been provided in the revised version. Lamps is a 200 W quartz tungsten halogen type operated at 3150oK.
- You need to correlate the mechanical characterization results with the elemental analysis and FTIR. Since the results obtained via elemental analysis and FTIR are limited, the correlation that you have shown in the manuscript may be improper or inaccurate.
Response: The authors agree. The results obtained via elemental analysis and FTIR are limited. However, as a first approximation, they are useful to investigate the possible correlation with other results concerning the polymeric samples as films as a whole.
Thank you very much for your comments and suggestions to improve presentation and content of the submitted paper. The revised version includes all these changes.
Round 2
Reviewer 1 Report
The changes are only cosmetic. The authors reply does not make any better conclusion. Required changes were not made.
The authors sould accept the fact thaht some parts of the manuscript are confusing. It is expected that your research is understanable to you since you have written it, but to someone who reads it for the first time, some questions occure. Thus you cannot claim the manuscript is not understandable, since it is.
lines 119-124, 132-140: are recognised in the plagiarism software as taken form the other source (the same one), the lines from 143-146 from the other source. This should be corrected.
missing the refrence for the calucaltion of absorbance in line 209.
The FTIR section is not covered with any refrence. Moreover you did not add the KBr tehnique into the experimental part. The experiment should be repetable.
The problem of averaged values is that - how can you average the values of different samples, for example the values of P13 and P16 into one?
lines 495-497: based on what parameters you know that this polymer is suitable for Mediterranean zone?
lines 531-541: how can you give a conslusion for what climatc zones the studied film were used? have you conducted the determination of the irradiance amaount in the named climatic zones
Author Response
It goes in attached document

Reviewer 4 Report
The authors have attempted to address the comments. But, I feel the response is incomplete. Some parts of the queries are unanswered. A detailed analysis of FTIR is required because the presence of other functional groups is detrimental to the mechanical properties/transmittance. Hence, it should be included in the revised version. Further, limiting the elemental analysis to only C, H & O doesn't make any sense and it cannot be considered as a first approximation. Hence, it is necessary to study the role of various elements by doing detailed elemental analysis and it has to be coupled with the characterization studies. I think the authors themselves have stressed the significance of filler elements in their introduction section itself. Hence, it cannot be discarded since these fillers hugely influence the mechanical and transmittance properties. Otherwise, the provided conclusions are disputable.
Author Response
It goes in attached document
